

# Production rate calibration for cosmogenic [10]Be in pyroxene by applying a rapid fusion method to [10]Be-saturated samples from the Transantarctic Mountains, Antarctica.

Marie Bergelin[1], Greg Balco[2,1], Lee B. Corbett[3] and Paul R. Bierman[3]

[1]Berkeley Geochronology Center, Berkeley, CA
[2]Lawrence Livermore National Lab, Livermore, CA
[3]Rubenstein School of the Environment and Natural Resources, University of Vermont/National Science Foundation Community Cosmogenic Facility.

*Correspondence to*: Marie Bergelin (mbergelin@bgc.org)

**Abstract**

Measurements of multiple cosmogenic nuclides in a single sample are valuable for various applications of cosmogenic nuclide exposure dating and allow for correcting exposure ages for surface weathering and erosion and establishing exposure-burial history. Here we provide advances in the measurement of cosmogenic [10]Be in pyroxene and constraints on the production rate which provide new opportunities for measurements of multi-nuclide systems, such as [10]Be/[3]He, in

pyroxene-bearing samples. We extracted and measured cosmogenic [10]Be in pyroxene from two sets of Ferrar Dolerite samples collected from the Transantarctic Mountains in Antarctica. One set of samples has [10]Be concentrations close to saturation which allows for the production rate calibration of [10]Be in pyroxene by assuming production-erosion equilibrium. The other set of samples, which has a more recent exposure history, is used to determine if a rapid fusion method can be successfully applied to samples with Holocene to Last-Glacial-Maximum exposure ages. From measured [10]Be concentrations

in the near-saturation sample set we find the production rate of [10]Be in pyroxene to be 3.74 +/- 0.10 atoms g[-1] yr[-1] and is consistent with [10]Be/[3]He paired nuclide ratios from samples assumed to have simple exposure. Given the high [10]Be concentration measured in this sample set, a sample mass of ~0.5 g of pyroxene is sufficient for the extraction of cosmogenic [10]Be from pyroxene using a rapid fusion method. However, for the set of samples having low [10]Be concentrations, measured concentrations were higher than expected. We attribute spuriously high [10]Be concentration to potential failure in removing

all meteoric [10]Be and/or a highly variable and poorly quantified measurement background.

## 1 Introduction

This paper describes advances in the measurement and application of cosmogenic [10]Be in pyroxene, including a rapid fusion extraction method and a production rate calibration data set. This is important because measurements of multiple cosmogenic nuclides in single samples are valuable for various applications of exposure dating. Multiple-nuclide systematics



are useful for correcting exposure ages for surface weathering and erosion (Klein et al., 1986; Nishiizumi et al., 1986; Lal, 1991), as well as quantifying when and how often a surface has experienced burial (Granger and Muzikar, 2001; Granger, 2006; Balco and Rovey, 2008). For quartz-rich samples, paired $^{26}Al/^{10}Be/^{21}Ne$ measurements in quartz are common practice and well-established (e.g. Balco and Shuster, 2009). However, multiple-nuclide measurements are generally not feasible in minerals other than quartz.


The stable cosmogenic nuclide $^3He$ is most commonly used in mafic rocks for exposure dating, as it is retentive in both pyroxene and olivine (Blard, 2021) and easily measured using a noble gas mass spectrometer (Balter-Kennedy et al., 2020). Measurements of cosmogenic $^{10}Be$ in pyroxene are potentially useful for exposure age applications and have been investigated in prior studies (Balter-Kennedy et al., 2020; Blard et al., 2008; Collins, 2015; Eaves et al., 2018; Ivy-Ochs et

al., 1998; Nishiizumi et al., 1990). To fully utilize multiple nuclides in pyroxene, it is necessary to constrain the production rate of cosmogenic $^{10}Be$ in pyroxene.

Cosmogenic nuclide production rates can be quantified in samples by (i) constraining the exposure age by independent radiocarbon and/or other geological dating methods (e.g. Borchers et al., 2016; Blard et al., 2008; Eaves et al., 2018), (ii)

measuring the ratio of one nuclide to another with an already well-known production rate(e.g. Niedermann et al., 1994), and/or (iii) samples experiencing negligible erosion rates and where the nuclide concentration has reached production-erosion equilibrium (Borchers et al., 2016; Jull et al., 1989; Nishiizumi et al., 1986). In this study, we take advantage of some of the longest exposed rocks in central Antarctica, where erosion rates are negligible, and $^3He$ exposure ages exceeding 8 Ma require that $^{10}Be$ concentrations must be close to the production-erosion equilibrium (Balter-Kennedy et al., 2020).

This provides an opportunity to validate the previously suggested $^{10}Be$ production rate in pyroxene constrained by the different approaches described above.

Previously, extraction of $^{10}Be$ from pyroxene (e.g. Balter-Kennedy et al., 2020; Blard et al., 2008; Collins, 2015; Eaves et al., 2018) has used wet chemical dissolution and column chromatography similar to that for extracting $^{10}Be$ from quartz (Corbett

et al., 2016). However, this process is challenging because of the large cation load and the extremely high selectivity required in the column separation. We adopt a $^{10}Be$ extraction method involving a total rapid fusion of the pyroxene sample (Stone, 1998) to improve the efficiency of $^{10}Be$ extraction from pyroxene. This method is commonly used to extract meteoric $^{10}Be$ from a variety of geologic matrices and should therefore be applicable for pyroxene despite the high concentrations of other cations.


We apply the fusion method to two sets of samples. First, we analyze a set of samples with extremely high $^{10}Be$ concentrations ($10^7$ atoms g$^{-1}$) that, as described above, can be used for production rate calibration by assuming production-





erosion equilibrium. Second, we analyze an additional set of samples with much lower $^{10}$Be concentrations ($10^4$-$10^5$ atoms g$^{-1}$) to determine if the fusion method can be successfully applied to samples with Holocene to Last-Glacial-Maximum exposure ages.

## 2 Method

### 2.1 Geological Setting and Samples

We selected two sets of samples of Ferrar Dolerite from the Transantarctic Mountains (TAM). The Ferrar Dolerite (Harvey, 2001) is a mafic intrusive rock consisting primarily of calcic plagioclase and several ortho- and clinopyroxenes (Elliot and Fleming, 2021). The first set consists of 10 samples from the upper TAM that had previous $^3$He measurements indicating exposure ages > 8 Ma. These samples are surface boulders collected from various moraines from Roberts Massif described by Balter-Kennedy et al. (2020) and several similar samples from nearby Otway Massif (Table 1). The Otway Massif data are not described in a publication but are available in the ICE-D database ([www.ice-d.org](www.ice-d.org)) due to public release requirements of the funding agency. Erosion rates for Ferrar Dolerite in Antarctica are 0-35 cm Myr$^{-1}$ (Balter-Kennedy et al., 2020). However, the $^3$He exposure ages limit the erosion rates for these specific samples to be < 5 cm yr$^{-1}$, and therefore, this set of samples can be expected to have reached production-decay equilibrium ("saturation") for $^{10}$Be, such that $N_{10} = P_{10}/\lambda_{10}$. After 8 Ma of exposure, $^{10}$Be concentrations have reached 98% of saturation values. Thus, these samples are expected to have extremely high $^{10}$Be concentrations, facilitating precise measurements. Measuring $^{10}$Be in these samples allows a straightforward estimate of the $^{10}$Be production rate in pyroxene integrated over the last 8 Ma.

The second set of samples is designed to test whether or not the fusion extraction method is also effective for samples with lower $^{10}$Be concentrations. The samples we analyze are low-elevation glacially transported erratics near outlet glaciers of the East Antarctic Ice Sheet in Northern Victoria Land. Exposure-age chronologies using $^{10}$Be in quartz or $^3$He in pyroxene from the same sites indicate that these samples have exposure ages of the last glacial-interglacial cycle. In addition, $^{10}$Be in pyroxene was previously measured in two of these samples (MG-12 and MG-19) using a dissolution/cation exchange method by Eaves et al. (2018). We selected this set of samples in part because they had been analyzed for $^3$He in previous studies (Table 1). We made several additional $^3$He measurements so that the entire sample set now has both $^3$He and $^{10}$Be data. The $^3$He data provide a means of evaluating the accuracy of the $^{10}$Be measurements. Details of the previously analyzed samples are from Stutz et al. (2021) and Eaves et al. (2018) and are summarized in Table 1.



**Table 1. Location and site information for samples of Ferrar Dolerite analyzed in this study.**

| Sample ID | Location | Latitude (Degrees) | Longitude (Degrees) | Elevation (m) | Thickness (cm) | Shielding | Prior Publication |
|---|---|---|---|---|---|---|---|
| 15-ROB-07 | Roberts Massif | -85.5249 | -177.7249 | 2255 | 2.0 | 0.9939 | Balter-Kennedy et al., (2020) |
| 15-ROB-27 | Roberts Massif | -85.5219 | -177.7279 | 2247 | 4.8 | 0.9959 | Balter-Kennedy et al., (2020) |
| 15-ROB-30 | Roberts Massif | -85.5101 | -177.7943 | 2385 | 4.4 | 1.0000 | Balter-Kennedy et al., (2020) |
| 15-ROB-31 | Roberts Massif | -85.5090 | -177.7788 | 2369 | 4.3 | 1.0000 | Balter-Kennedy et al., (2020) |
| 15-OTW-50 | Otway Massif | -85.4159 | 172.8086 | 2268 | 1.4 | 0.9967 | Unpublished |
| 15-OTW-55 | Otway Massif | -85.4150 | 172.7819 | 2292 | 2.7 | 0.9962 | Unpublished |
| 15-OTW-56 | Otway Massif | -85.4146 | 172.7756 | 2290 | 3.1 | 0.9959 | Unpublished |
| 15-OTW-57 | Otway Massif | -85.4148 | 172.7832 | 2287 | 1.3 | 0.9962 | Unpublished |
| 15-OTW-58 | Otway Massif | -85.4371 | 172.8626 | 2504 | 2.0 | 0.9980 | Unpublished |
| 15-OTW-60 | Otway Massif | -85.4370 | 172.8670 | 2503 | 1.8 | 0.9980 | Unpublished |
| 17-HB-TC-02 | Hughes Bluff | -75.3918 | 162.2125 | 120.8 | 1.0 | 0.9962 | Stutz et al., (2021) |
| 17-HB-TC-12 | Hughes Bluff | -75.3957 | 162.2021 | 185.3 | 1.0 | 0.9919 | Stutz et al., (2021) |
| 17-EHW-05 | Evans Heights | -75.0982 | 161.4989 | 433 | 1.0 | 1.0000 | Stutz et al., (2021) |
| 17-EHW-15 | Evans Heights | -75.0947 | 161.4969 | 561 | 1.0 | 1.0000 | Stutz et al., (2021) |
| 15-MG12 | MacKay Gl. | -76.9985 | 161.0376 | 1013 | 5.8 | 0.9790 | Eaves et al., (2018) |
| 15-MG19 | MacKay Gl. | -76.9991 | 161.0406 | 981 | 4.0 | 0.9880 | Eaves et al., (2018) |

## 2.2 Mineral Separation

The samples were crushed and sieved to a grain size of 75-125 $\mu$m at which mostly monomineralic grains were observed. The samples were washed in water and then leached in 10% HCl at room temperature overnight. We then ran the sample through a magnetic separator to separate pyroxene from the less magnetic plagioclase and other minerals present.

At the National Science Foundation / University of Vermont Community Cosmogenic Facility (CCF), the pyroxene grains underwent HF leaching, following Balter-Kennedy et al. (2023), to remove meteoric [10]Be and any plagioclase attached to the pyroxene grains. A fine grain size reduces the amount of meteoric [10]Be stored in the grain fractures, and HF etching was found to be sufficient to remove meteoric [10]Be by Balter-Kennedy et al (2023), without powdering the sample as otherwise previously suggested (Blard et al., 2008). The samples were leached in HF twice; first in a solution of 1% HF in an ultrasonic bath at ~60 °C for 6 hours and then again in 1% HF/1% $HNO_3$ overnight, targeting a 20-30 % mass loss. During





HF leaching, precipitates of fluoride ($MgF_2$, $CaF_2$) are produced and are insoluble in dilute HF. Therefore, we did a final leaching in 0.5% $HNO_3$ overnight in a heated ultrasonic bath to dissolve the fluoride precipitates.

## 2.3 Extraction and analyses of cosmogenic [10]Be in Pyroxene

The extraction of Be was done at the CCF by total fusion in a potassium bifluoride ($KHF_2$) flux according to Stone (1998).
Samples were processed in two separate batches; the first batch contained the high-concentration samples, and the second batch contained the low-concentration samples. The pure pyroxene samples were powdered using a shatterbox, and 0.5 g of powdered sample was massed into 30 mL platinum crucibles. The sample mass is determined by the size of the Pt crucibles and other properties of the heating apparatus and is chosen to avoid spattering and sample loss during fusion. For the set of samples with expected high [10]Be concentration, we added 400 $\mu$g of [9]Be carrier to each 0.5-g sample. This [9]Be carrier is a
beryl carrier (termed Carrier C) made at the facility with a concentration of 348 $\mu$g/mL. After drying the sample and carrier mixture, anhydrous $KHF_2$ and anhydrous $Na_2SO_4$ were added at the ratio of 8:1:2 $KHF_2$:$Na_2SO_4$:sample by weight to the crucibles and homogenized.

The fusion protocol at the University of Vermont uses 30 mL platinum crucibles which for safety reasons and splatter
control, limits sample size to 0.5 g. While it is possible to fuse larger (1-2 g) samples in larger (100 ml) crucibles (Stone, 1998), these are not compatible with the fixed fluxing apparatus used to minimize the hazard of molten $KHF_2$. To increase the sample size and the measured [10]Be/[9]Be ratio for the set of expected low [10]Be concentration samples, we fused 1 g of sample in two separate fusions of 0.5 g each, with half as much carrier (200 $\mu$g) as used for the initial sample batch. With sample and carrier concentrations similar in both aliquots (specifically, as close as possible with the weighing and dispensing
equipment in use; we estimate better than 1% agreement between aliquots), [10]Be/[9]Be ratios in both aliquots after fusion can be expected to be identical, so we combined them to yield a higher sample/carrier ratio than possible in a single fusion.

Before starting this procedure, we determined whether halving the amount of [9]Be carrier would affect the Be yield, by fusing aliquots of sample 15-OTW-60 with varying amounts of added [9]Be carrier. Total [9]Be yields (Table 2) show that less [9]Be
does not result in a lower Be yield. Because Be yields in the first set of samples were lower than expected, we increased the amount of $Na_2SO_4$ added to a ratio of 4:2:1 $KHF_2$:$Na_2SO_4$:sample by weight as suggested for calcium-rich samples by Stone (1998). This change makes sense because the Ferrar pyroxene is calcic; having an abundance of $SO_4$ during fluxing suppresses the formation of $CaBeF_4$, which is less soluble. This modification significantly increased the total Be yield (Table 2).


After fusion, the Stone (1998) procedure involves Be and K extraction by water leaching, and removal of residual fluorides by centrifuging. At this point, the two aliquots of each sample were combined, and K was removed from the combined



sample by precipitation of $KClO_4$, evaporation of the supernatant to remove the remaining $HClO_4$, and redissolution in 12 mL of dilute $HNO_3$.


At this point, we experienced difficulty in completely redissolving the precipitated sample and found it necessary to centrifuge the sample multiple times to remove what we presumed to be the remaining $KClO_4$. Although Be yields from these samples were as expected (Table 2), the resulting AMS targets had unusually low beam currents (given the fraction of beam current of other samples), which made AMS measurement more difficult than expected. We hypothesize that this is

most likely the result of K carryover in the final stages of the extraction process and that this could have been prevented by increasing the volume of the final $HNO_3$ solutions to dissolve K more effectively.

Ratios of $^{10}Be/^9Be$ were measured at Lawrence Livermore National Laboratory (LLNL) and normalized to the 07KNSTD3110 standard (Nishiizumi et al., 2007) with a $^{10}Be/^9Be$ ratio of 2.85 x $10^{-12}$. Uncertainties in calculated $^{10}Be$

concentrations include AMS measurement uncertainties, uncertainty on the Be carrier concentration, and uncertainty in blank corrections (Table 2). Five procedural blanks measured with both sample batches had a mean and standard deviation of $128000 \pm 67000$ atoms $^{10}Be$. This is less than 0.4% of the total amount of $^{10}Be$ measured in any of the samples in the high-concentration batch (Table 2), so blank correction uncertainty makes a negligible contribution to overall measurement uncertainty for these samples. However, the highest blank values were up to 60% of the total number of atoms measured in

some of the low-concentration samples, so blank uncertainty is significant for the low-concentration batch. We discuss this in more detail in section 3.5.

### 2.4 Cosmogenic $^3He$ Analysis

We measured cosmogenic $^3He$ concentrations in all samples at Berkeley Geochronology Center (BGC) following the procedure described in Balter-Kennedy et al. (2020). $^3He$ concentrations for two samples, HB-TC-02 and HB-TC-12, have

already been reported in Stutz et al.(2021). Measurements of the CRONUS-P intercomparison standard (Blard et al., 2015) during the period of these measurements were $5.03 \pm 0.15$ x $10^9$ atoms $g^{-1}$ $^3He$ (Balter-Kennedy et al., 2020), which is indistinguishable from the accepted value of $5.02 \pm 0.12$ x $10^9$ atoms $g^{-1}$ (Blard et al., 2015).

### 3 Results and Discussion

### 3.1 Measured cosmogenic $^{10}Be$ in saturated samples.

Measured $^{10}Be$ concentrations in the set of high-concentration samples range from $5.92 - 7.67$ x $10^7$ atoms $g^{-1}$ with uncertainties < 2.2 % (Tables 2 and 3). These are equivalent to some of the highest $^{10}Be$ concentrations measured in terrestrial rocks (Spector and Balco, 2020). As expected from the elevation dependence of the $^{10}Be$ production rate and the





assumption that the $^{10}$Be concentrations are close to production-decay saturation, the measured concentrations increase systematically with elevation (Fig. 1).




**Table 2 Measure Be results, including yields measured by ICP-OES in the dilute HNO3 solution prior to final precipitation, with implied Be yields for the fusion process and measured AMS current and ratios.**

| Sample Name | Pyroxene Mass (g) | 9Be Added (µg) | Be Yield (µg) | Be Yield (%) | AMS 10Be/9Be | Mean 9Be current Relative to standard[a] | Measured 10Be (10^6 atoms) | Blank corr. 10Be Conc. (10^6 atoms g^-1) |
|---|---|---|---|---|---|---|---|---|
| *High-concentration batch* | | | | | | | | |
| 15-ROB-07 | 0.493 | 403 | 110 | 27 | 1.281 ± 0.024 x 10^-12 | 0.48 | 34.89 ± 0.75 | 70.5 ± 1.5 |
| 15-ROB-27 | 0.497 | 403 | 118 | 29 | 1.085 ± 0.018 x 10^-12 | 0.54 | 29.57 ± 0.56 | 59.2 ± 1.1 |
| 15-ROB-30 | 0.488 | 402 | 145 | 36 | 1.222 ± 0.023 x 10^-12 | 0.55 | 33.21 ± 0.7 | 67.8 ± 1.4 |
| 15-ROB-31 | 0.501 | 400 | 132 | 33 | 1.192 ± 0.018 x 10^-12 | 0.66 | 32.21 ± 0.59 | 64 ± 1.2 |
| 15-OTW-50 | 0.498 | 398 | 117 | 30 | 1.165 ± 0.022 x 10^-12 | 0.59 | 31.34 ± 0.67 | 62.7 ± 1.3 |
| 15-OTW-55 | 0.496 | 402 | 117 | 29 | 1.139 ± 0.021 x 10^-12 | 0.47 | 30.96 ± 0.66 | 62.2 ± 1.3 |
| 15-OTW-56 | 0.498 | 399 | 108 | 27 | 1.232 ± 0.023 x 10^-12 | 0.53 | 33.23 ± 0.7 | 66.5 ± 1.4 |
| 15-OTW-57 | 0.490 | 397 | 113 | 28 | 1.182 ± 0.022 x 10^-12 | 0.60 | 31.71 ± 0.67 | 64.5 ± 1.4 |
| 15-OTW-58 | 0.501 | 399 | 107 | 27 | 1.429 ± 0.028 x 10^-12 | 0.50 | 38.56 ± 0.85 | 76.7 ± 1.7 |
| 15-OTW-60 | 0.497 | 398 | 114 | 29 | 1.369 ± 0.026 x 10^-12 | 0.47 | 36.87 ± 0.78 | 73.9 ± 1.6 |
| 15-OTW-60-150[b] | 0.493 | 159 | 64 | 40 | - | - | | |
| 15-OTW-60-250[b] | 0.495 | 258 | 79 | 31 | - | - | | |
| Blank (129-BLK) | - | 398 | 279 | 70 | 5.1 ± 1 x 10^-15 | 0.80 | 0.139 ± 0.028 | |
| Blank (129-BLKX) | - | 404 | 267 | 66 | 5.28 ± 0.48 x 10^-15 | 0.62 | 0.144 ± 0.013 | |
| Blank (129-0BLK) | - | 402 | 297 | 74 | 2.18 ± 0.27 x 10^-15 | 0.79 | 0.0594 ± 0.0074 | |
| *Low-concentration batch* | | | | | | | | |
| 17-HB-TC-02 | 0.998 | 400 | 268 | 67 | 2.53 ± 0.11 x 10^-14 | 0.49 | 0.685 ± 0.03 | 0.558 ± 0.074 |
| 17-HB-TC-12 | 0.997 | 400 | 250 | 63 | 2.03 ± 0.11 x 10^-14 | 0.36 | 0.55 ± 0.03 | 0.424 ± 0.074 |
| 17-EHW-05 | 0.998 | 399 | 242 | 61 | 1.67 ± 0.13 x 10^-14 | 0.22 | 0.451 ± 0.034 | 0.323 ± 0.075 |
| 17-EHW-15 | 0.999 | 399 | 267 | 67 | 3.70 ± 0.17 x 10^-14 | 0.27 | 0.997 ± 0.046 | 0.87 ± 0.082 |
| 15-MG12 | 1.001 | 398 | 281 | 71 | 2.40 ± 0.13 x 10^-14 | 0.32 | 0.646 ± 0.037 | 0.517 ± 0.076 |
| 15-MG19 | 1.000 | 399 | 263 | 66 | 3.96 ± 0.55 x 10^-14 | 0.10 | 1.07 ± 0.15 | 0.94 ± 0.16 |
| Blank (130-BLK) | - | 399 | 333 | 83 | 8.3 ± 1.2 x 10^-15 | 0.17 | 0.226 ± 0.032 | |
| Blank (130-BLKX) | - | 399 | 333 | 83 | 2.62 ± 0.54 x 10^-15 | 0.25 | 0.071 ± 0.015 | |

[a] Mean current for the KNSTD3110 is 21.5 µA
[b] Sample were processes only as a yield test and no AMS measurements were made






**Table 3. $^3$He and $^{10}$Be concentrations for long-exposed glacial erratics in the Transantarctic Mountains. The $^{10}$Be production rate is determined from Eq. 1.**

| Sample ID | $^{10}$Be conc. ($10^9$ atoms g) | $^3$He conc. ($10^9$ atoms g) | $^3$He exposure age (Myrs) | $^{10}$Be production rate SLHL spallation[a] (atoms g$^{-1}$ yr$^{-1}$) | $^3$He data from |
|---|---|---|---|---|---|
| 15-ROB-07 | 7.05 ± 0.15 | 9.19 ± 0.18 | 8.12 ± 0.16 | 4.26 | Balter-Kennedy et al., (2020) |
| 15-ROB-27 | 5.92 ± 0.11 | 9.05 ± 0.10 | 8.265 ± 0.094 | 3.69 | Balter-Kennedy et al., (2020) |
| 15-ROB-30 | 6.78 ± 0.14 | 12.21 ± 0.35 | 9.95 ± 0.29 | 3.78 | Balter-Kennedy et al., (2020) |
| 15-ROB-31 | 6.40 ± 0.12 | 10.51 ± 0.14 | 8.67 ± 0.12 | 3.62 | Balter-Kennedy et al., (2020) |
| 15-OTW-50 | 6.27 ± 0.13 | 10.84 ± 0.26 | 9.40 ± 0.23 | 3.68 | ICE-D Database[b] |
| 15-OTW-55 | 6.22 ± 0.13 | 11.07 ± 0.13 | 9.56 ± 0.11 | 3.64 | ICE-D Database[c] |
| 15-OTW-56 | 6.65 ± 0.14 | 10.53 ± 0.13 | 9.14 ± 0.12 | 3.92 | ICE-D Database[c] |
| 15-OTW-57 | 6.45 ± 0.14 | 10.87 ± 0.16 | 9.28 ± 0.13 | 3.74 | ICE-D Database[c] |
| 15-OTW-58 | 7.67 ± 0.17 | 12.4235 ± 0.0092 | 9.0549 ± 0.0067 | 3.88 | ICE-D Database[c] |
| 15-OTW-60 | 7.39 ± 0.16 | 11.73 ± 0.23 | 8.54 ± 0.17 | 3.74 | ICE-D Database[c] |

[a] The reference 10Be production rate is determined from Equation (1) and the scaling method of Stone (2000), as implemented Balco et al., (2008)
[b] https://version2.ice-d.org/antarctica/site/CHARLIE/
[c] https://version2.ice-d.org/antarctica/site/OTWEBAS/




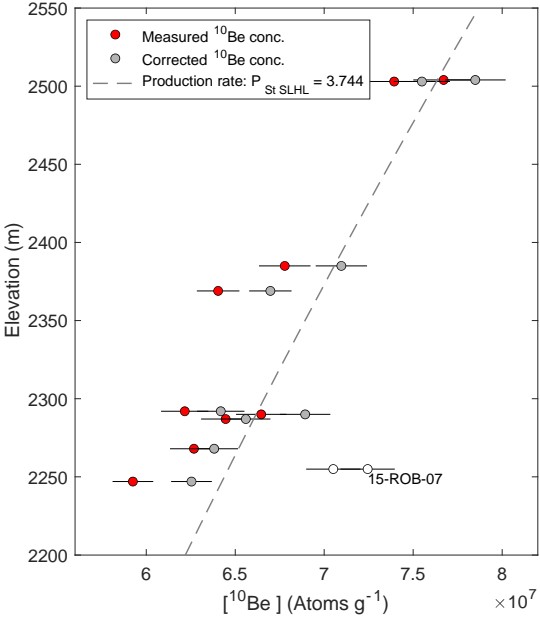

**Figure 1: Measured $^{10}$Be concentrations versus elevation. Red dots are measured $^{10}$Be concentrations as reported in Table 3, gray dots show measured $^{10}$Be concentrations corrected for sample thickness and shielding, and dashed line show the saturated $^{10}$Be concentrations for the 'St' reference production rate of 3.74 atoms g$^{-1}$ yr$^{-1}$ $^{10}$Be in pyroxene. White dots indicate sample outlier,**
**which is not included in the production rate calibration (see section 3.2).**

### 3.2 $^{10}$Be Production rate in pyroxene

In general, as discussed above, $^{3}$He exposure ages range between 8-10 Ma (5-6 times the $^{10}$Be half-life) and imply that $^{10}$Be concentrations in these samples are within 1-2% of production-decay saturation. We account for the small, predicted difference from the saturation concentration by calculating the production rate as,


$$P_{10} = \frac{N_{10}\,\lambda_{10}}{(1 - e^{-\lambda_{10}t_3})}\,, \tag{1}$$

where $N_{10}$ is the $^{10}$Be concentration (atoms g$^{-1}$), $P_{10}$ is the $^{10}$Be production rate in the sample (atoms g$^{-1}$yr$^{-1}$), $\lambda_{10}$ is the $^{10}$Be decay constant (4.99 x 10$^{-7}$ yr$^{-1}$), and $t_3$ is the $^{3}$He exposure age (yr). Because the samples are close to production-decay
saturation, the production rate determined from Eq. 1 is insensitive to uncertainty in the assumed exposure age. Therefore, although we use the apparent $^{3}$He exposure ages to correct for an inferred small systematic difference from production-decay saturation, the accuracy of the $^{3}$He ages is minimally important for the $^{10}$Be production rate estimate. To obtain the



spallogenic production rate of $^{10}$Be in pyroxene, we subtract the production rate in pyroxene due to muons using the muon interaction cross-sections of Balter-Kennedy et al. (2023) and correct for sample thickness and topographic shielding.


Applying the 'St' elevation scaling of Stone (2000) then yields sea level/high latitude (SLHL) production rates in the range of 3.5-4.1 atoms g$^{-1}$ yr$^{-1}$ (Table 3). The $^{10}$Be production rate increases with elevation, so samples near or at saturations are expected to likewise have $^{10}$Be concentrations increase with elevation. This is true for all samples, except 15-ROB-07, which have an excess $^{10}$Be concentrations equivalent to ~250 m (Fig. 1). Removing one outlier (15-ROB-07, see Fig. 1) yields a

mean and standard deviation of 3.74 ± 0.10 atoms g$^{-1}$ yr$^{-1}$.

The production rate estimate agrees with that of Balter-Kennedy et al. (2023) (3.6 ± 0.2 atoms g$^{-1}$ yr$^{-1}$). However, in this study, samples with near-saturated $^{10}$Be concentrations permit a direct calculation of the production rate from the measurements. In contrast, the sample set in the Balter-Kennedy et al. (2023) study lacks direct constraints on the exposure

age and/or exposure history, and a best-fit production rate was computed from values that permitted all the samples to have a simple exposure history bounded by limiting assumptions of steady exposure at zero erosion and steady erosion for an infinite time. While they are not directly comparable, it is possible to determine whether the two data sets are consistent with each other and with the assumption of simple exposure. In Fig. 2 we construct a $^{10}$Be/$^{3}$He two-nuclide diagram using the production rate determined from our study and plot the $^{10}$Be/$^{3}$He data from both studies. This shows that all data from both

studies (except for one outlier in our study identified above) plot within the simple exposure region and are therefore internally consistent.



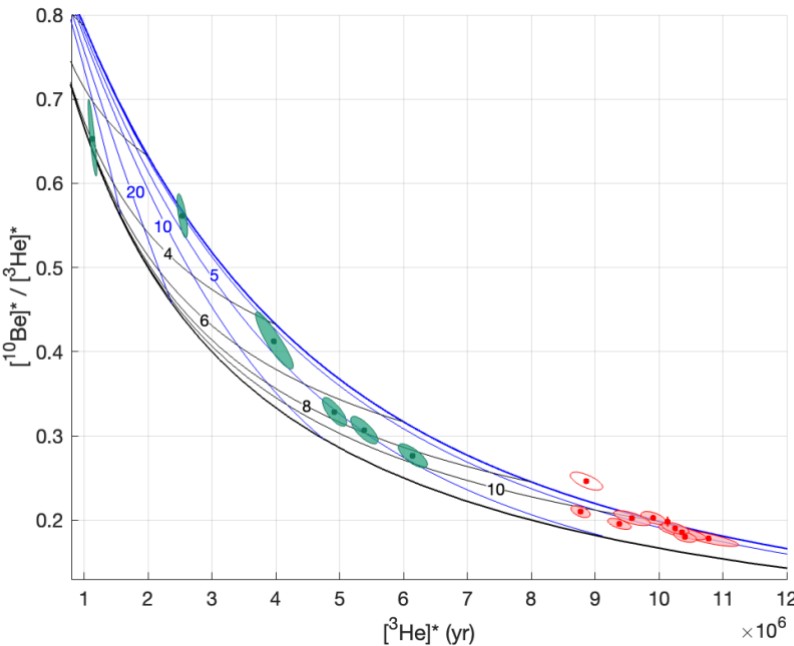

**Figure 2:** **$^{10}$Be-$^3$He two-nuclide diagram. Red data points show measurements from this study, green data are from Balter-Kennedy et al., (2023), where each shaded ellipse represents the 68% confidence interval in the measured uncertainty. Thick blue line is the simple exposure line and the thin blue lines are lines of constant erosion (m Myr$^{-1}$). Thick black is the steady-erosion line, and the thin black lines are constant age (Myr). * signifies nuclide concentrations normalized to site-specific production rate for comparison across sampling locations.**

Finally, we consider whether our data are consistent with other $^{10}$Be-in-pyroxene production rate calibration data and with commonly used production rate scaling methods. Two other studies obtained $^{10}$Be-in-pyroxene production rate calibration data from samples with independent age constraints. Blard et al. (2008) included two samples (SI41 and SI43) from separate lava flows at Mt. Etna, Italy with K/Ar ages of 33 kyrs and 10 kyrs, respectively. Eaves et al. (2018) obtained three samples from the Murimotu formation debris avalanche at Mt Ruapehu, New Zealand, which has a radiocarbon age of 10.5 kyrs.

In Fig. 3, we apply the production rate calibration code from version 3 of the online exposure age calculator originally described by Balco et al. (2008) and subsequently updated, to (i) our production rate calibration data alone, and (ii) our data with the Blard et al. (2008) and Eaves et al. (2018) data. One potentially important aspect of this comparison is that our data are from relatively high elevations and high latitudes, and the other calibration data are from relatively low elevations and moderate latitudes. Therefore, this comparison is a potential test of the hypothesis that the time-dependent 'LSDn' scaling method (Lifton et al., 2014; Lifton, 2016) more accurately represents the elevation dependence of the production rate at high latitudes (Balco, 2016). In fact, Fig. 3 shows that, in agreement with this hypothesis, LSDn scaling suppresses an elevation-dependent residual in reference production rates calculated with the 'St' and 'Lm' scaling methods.




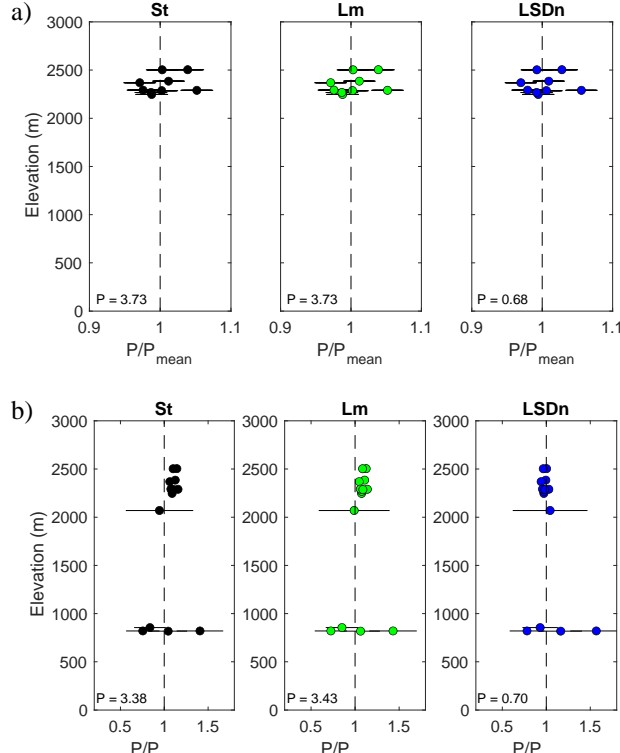


**Figure 3: Relative variation with elevation in production rate scaling parameters calculated from calibration samples in this study (high-elevation data; shown in both panels (a) and (b)) and those of Blard et al. (2008) and Eaves et al. (2018) (lower-elevation data; shown in panel (b) only). For the St and Lm scaling methods, the production rate scaling parameter P is a reference production rate with units of atoms g$^{-1}$ yr$^{-1}$; for the LSDn scaling method, it is a nondimensional correction factor. An elevation-**
**dependent residual is evident for St and Lm scaling but is resolved by LSDn scaling. This implies that LSDn scaling better represents the elevation dependence of the production rate at polar latitudes.**

Taken all together, we find that the reference production rate of 3.74 ± 0.10 atoms g$^{-1}$ yr$^{-1}$ determined in this study is in agreement with previously published production rates of 3.6 ± 0.2 atoms g$^{-1}$ yr$^{-1}$ with an overall improvement in the uncertainty.

**3.3 $^{10}$Be and $^{3}$He measurements in low-concentration samples**

The $^{10}$Be concentrations from the set of young-exposure-age erratics, as expected, were two orders of magnitude lower than concentrations in the high-elevation, saturated samples (Table 4). As discussed above, these samples are glacially transported erratics found near the margins of major glaciers in the Transantarctic Mountains. The geomorphic context, $^{3}$He exposure ages on these and nearby samples, and $^{10}$Be exposure ages on nearby quartz-bearing samples, all indicate that these samples



were emplaced by deglaciation during the last glacial-interglacial cycle and have most likely not experienced more than 50,000 years of exposure (Stutz et al., 2021; Eaves et al., 2018).

Given the assumptions that (i) the samples have experienced exposure only in the last ~50,000 years and (ii) the non-cosmogenic $^3$He concentration is constant among samples, measured $^3$He and $^{10}$Be concentrations should be linearly related,
with a slope given by the $^3$He/$^{10}$Be production ratio and an intercept on the $^3$He axis given by the non-cosmogenic $^3$He concentration in Ferrar pyroxene. Non-cosmogenic $^3$He in Ferrar pyroxene is most likely derived from nucleogenic production and has been estimated in various studies to be less than approximately 6 Matoms g$^{-1}$ (Ackert, 2000; Kaplan et al., 2017; Margerison et al., 2005).

Combining our $^3$He measurements with the $^{10}$Be concentrations obtained from Collins (2015) and Eaves et al. (2018) results in the expected linear relationship, with a slope of $^3$He/$^{10}$Be = 28.5 ± 4.6 and $^3$He intercept of 3.9 ± 0.8 x 10$^6$ atoms g$^{-1}$. If we take the reference $^3$He production rate to be 120 ± 13 atoms g$^{-1}$ yr$^{-1}$, which is derived for 'St' scaling with the calibration data set of Borchers et al., (2016), this slope implies a $^{10}$Be production rate of 4.20 ± 0.82 atoms g$^{-1}$ yr$^{-1}$, which is consistent with, although less precise than, the other estimates discussed in the previous sections. The $^3$He intercept is most likely a good
estimate of the nucleogenic $^3$He concentration in Ferrar pyroxene (Balco, 2020).

However, only one of the $^{10}$Be concentrations measured in this study agrees with the expected linear relationship; the others are systematically higher than expected, by hundreds of thousands of atoms g$^{-1}$. In particular, MG12 and MG19 were measured both by Eaves et al., (2018) and in this study; our results are 3.94 x 10$^5$ and 8.4 x 10$^5$ atoms g$^{-1}$ higher than the
Eaves et al., (2018) results, respectively (Table 4). Two possible explanations for this discrepancy are (i) failure to completely remove meteoric $^{10}$Be$_m$ before extraction, or (ii) a highly variable and poorly quantified measurement background (Table 2). Both scenarios are discussed in the following sections.



**Table 4 Measured ³He and ¹⁰Be concentrations in low-concentration samples from glacial transported erratics during the last glacial-interglacial cycle, including published concentrations from others.**

| Sample name | aliquot | Mass (g) | Measured $^4$He ($10^9$ atoms g$^{-1}$) | Total Measured $^3$He ($10^6$ atoms g$^{-1}$) | Total $^3$He weighted mean ($10^6$ atoms g$^{-1}$) | $^3$He data source | Measured $^{10}$Be ($10^6$ atoms g$^{-1}$) | $^{10}$Be data source |
|---|---|---|---|---|---|---|---|---|
| *Mt. Gran (Mackay Glacier)* | | | | | | | | |
| MG-01 | a | 0.03887 | 34.3 ± 1.2 | 5.88 ± 0.77 | 6.36 ± 0.42 | This paper | | |
| | b | 0.09641 | 35.3 ± 1.3 | 6.56 ± 0.50 | | | | |
| MG-02B | a | 0.04679 | 159.1 ± 5.7 | 8.40 ± 0.85 | 8.26 ± 0.48 | This paper | 0.055 ± 0.040 | Eaves et al. (2018) |
| | b | 0.08192 | 158.4 ± 5.6 | 8.15 ± 0.69 | | | | |
| | c | 0.04119 | 154.6 ± 5.6 | 8.31 ± 1.06 | | | | |
| MG-07 | a | 0.06049 | 34.7 ± 1.2 | 14.13 ± 0.80 | 14.13 ± 0.80 | This paper | 0.271 ± 0.062 | Collins (2015) |
| MG-08B | b | 0.01779 | 131.1 ± 4.7 | 22.27 ± 2.40 | 19.13 ± 1.12 | This paper | 0.337 ± 0.087 | Eaves et al. (2018) |
| | c | 0.04954 | 295.4 ± 10.6 | 18.26 ± 1.26 | | | | |
| MG-15 | a | 0.09931 | 84.9 ± 3 | 8.52 ± 0.63 | 7.77 ± 0.46 | This paper | 0.52 ± 0.10 | Collins (2015) |
| | b | 0.07935 | 81.3 ± 2.9 | 6.90 ± 0.67 | | | | |
| MG-22 | a | 0.09661 | 29.1 ± 1 | 7.34 ± 0.61 | 7.28 ± 0.53 | This paper | 0.182 ± 0.048 | Eaves et al. (2018) |
| | b | 0.03488 | 28.1 ± 1 | 7.10 ± 1.05 | | | | |
| MG-32 | a | 0.09666 | 36.5 ± 1.3 | 9.99 ± 0.62 | 9.54 ± 0.53 | This paper | 0.093 ± 0.036 | Eaves et al. (2018) |
| MG-12 | b | 0.03643 | 38 ± 1.4 | 8.34 ± 1.01 | 6.56 ± 1.02 | This paper | 0.135 ± 0.051 | Eaves et al. (2018) |
| | a | 0.02253 | 174.1 ± 1.5 | 7.29 ± 0.88 | | | | |
| | b | 0.01526 | 243.9 ± 2.1 | 5.40 ± 1.62 | | | | |
| | c | 0.02199 | 165.4 ± 1.4 | 6.98 ± 0.87 | | | | |
| MG-19 | a | 0.02329 | 583.7 ± 4.9 | 7.16 ± 1.02 | 7.78 ± 2.32 | This paper | 0.123 ± 0.034 | Eaves et al. (2018) |
| | | | | | | | 0.517 ± 0.076 | This paper |



| Sample | | | | | | | | |
|---|---|---|---|---|---|---|---|---|
| | c | 0.02600 | 590.9 ± 4.9 | 10.66 ± 1.05 | | Eaves et al. (2018) | 0.098 ± 0.054 | |
| | d | 0.01643 | 602.0 ± 4.9 | 6.02 ± 1.32 | | This paper | 0.94 ± 0.16 | |
| | e | 0.01431 | 525.2 ± 4.4 | 9.74 ± 1.66 | | | | |
| | f | 0.01403 | 490.1 ± 4.1 | 5.34 ± 1.50 | | | | |
| *Evans Heights (David Glacier)* | | | | | | | | |
| EHW-05 | a | 0.02364 | 108.6 ± 1.8 | 3.76 ± 1.67 | 2.91 ± 0.7 | This paper | | |
| | b | 0.06775 | 108.0 ± 1.9 | 4.43 ± 0.87 | | | | |
| | c | 0.05934 | 107.7 ± 1.9 | 1.60 ± 0.75 | | | | |
| EHW-15 | a | 0.02905 | 216.5 ± 3.7 | 6.91 ± 1.46 | 6.3 ± 1.1 | This paper | 0.323 ± 0.075 | |
| | b | 0.03577 | 179.9 ± 3.1 | 4.43 ± 1.40 | | | | |
| | c | 0.03328 | 178.3 ± 3.1 | 7.73 ± 1.52 | | | | |
| *Hughes Bluff (David Glacier)* | | | | | | | | |
| HB-TC-02 | a | 0.02268 | 230.0 ± 5.5 | 11.85 ± 2.08 | 8.8 ± 1.4 | Stutz et al. (2021) | 0.87 ± 0.082 | This paper |
| | b | 0.03491 | 195.9 ± 3.4 | 8.16 ± 1.73 | | | | |
| | c | 0.03291 | 178.9 ± 3.1 | 7.49 ± 1.67 | | | | |
| HB-TC-12 | c | 0.01439 | 99.2 ± 1.7 | 17.48 ± 3.31 | 17.5 ± 3.3 | Stutz et al. (2021) | 0.558 ± 0.074 | This paper |
| | | | | | | | 0.424 ± 0.074 | This paper |

Notes:
1. All $^3$He measurements employed the BGC "Ohio" NGMS system. Analytical methods are as described in Balter-Kennedy et al., (2020)
2. $^{10}$Be data from Eaves and Collins were originally normalized to the NIST SRM4325 standard with an assumed $^{10}$Be/$^9$Be ratio of $3 \times 10^{-11}$, and have been renormalized to the '07KNSTD' standardization of Nishiizumi et al., (2007).



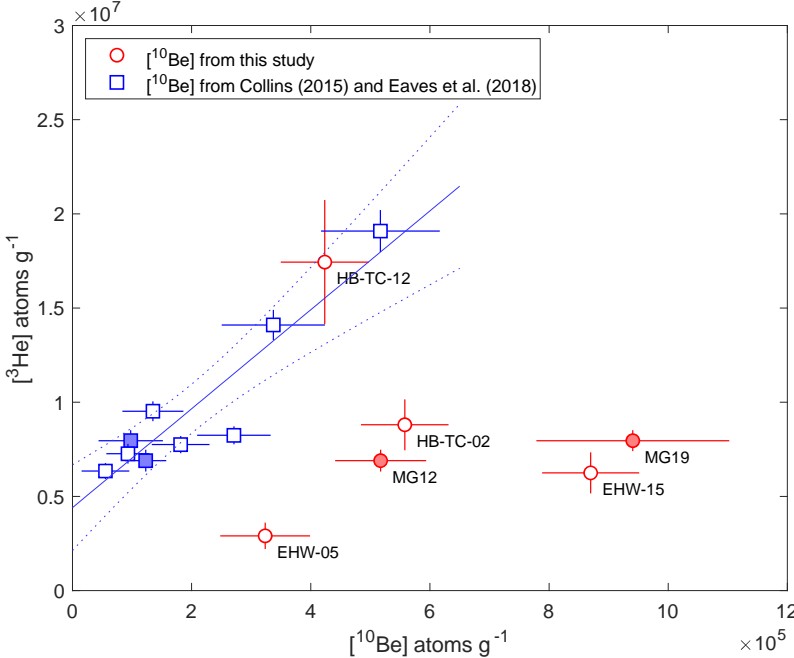

**Figure 4: Measured [10]Be and [3]He concentrations in low-concentration samples. Red dots are sample data with [10]Be concentrations measured in this study. Blue squares are sample data with [10]Be concentrations obtained from Collins (2016) and Eaves et al., (2018). Solid points represent samples having duplicated [10]Be measurements from this study and Eaves et al., (2018). The horizontal and vertical lines associated with each data point are the measured uncertainty in the nuclide concentrations. Blue solid line is the linear relationship for the blue data points only with a 95% confidence bound (dashed blue lines).**

### 3.4 Removal of meteoric $^{10}$Be

Failure to successfully remove all meteoric $^{10}$Be$_m$ during HF etching would result in spuriously high concentrations of presumed cosmogenic $^{10}$Be. Balter-Kennedy et al. (2023) found that when using fine to medium grains of pyroxene (32-125 $\mu$m), ~25% mass loss after leaching a sample in 1% HF/1%HNO$_3$ is sufficient to remove meteoric $^{10}$Be$_m$. After leaching, we observed 35–49% mass loss, indicating that leaching should have been sufficient. Figure 5 compares the mass lost during HF etching to the normalized residual between the measured and predicted cosmogenic $^{10}$Be concentration (atoms g$^{-1}$) calculated using the production rate from this study of 3.74 atoms g$^{-1}$ yr$^{-1}$ and the minimum $^3$He ages for both the high- and low-concentration samples. We see no clear relationship between mass loss and the $^{10}$Be residual for either of the two sample sets, as expected. This is especially evident in samples HB‑TC‑12 and MG19 which both display similar mass loss (~ 48 %).





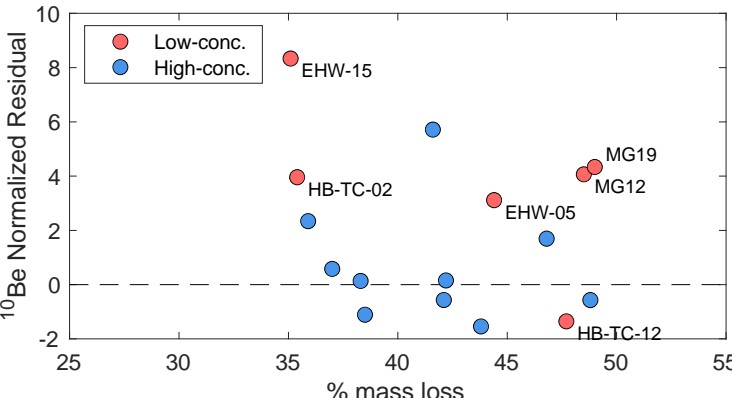

**Figure 5: Comparison of mass lost during HF etching prior to [10]Be extraction and the normalized residual between measured and**
**predicted cosmogenic [10]Be concentrations. Red data points are from the sample set of low [10]Be concentrations. Blue data points are**
**from the sample set having high [10]Be concentrations.**

If we were to assume that the increased [10]Be is solely meteoric, then that contributes $\sim 6 \times 10^5$ atoms g$^{-1}$, which is estimated
from the average difference between the [10]Be concentrations for the replicated samples. This would account for less than 1%
of the [10]Be concentration measured for the set of high-concentration samples used for estimating the production rate of [10]Be
in pyroxene. Therefore, any potential contribution from meteoric [10]Be would most likely have an insignificant impact on the
reference production rate reported in section 3.2.

As dissolved plagioclase attached to pyroxene grains contributes to the total mass loss after leaching, the total mass loss is
not a direct reflection of the mass of pyroxene lost and presumed to contain meteoric [10]Be$_m$. While the >35% mass loss is
mostly pyroxene, some unknown fraction could be from plagioclase. We can therefore not exclude that samples may contain
some meteoric [10]Be$_m$. However, the lack of correlation between the residuals vs. expected values and the mass loss during
etching makes it unlikely that the systematically measured increase in [10]Be concentration is solely caused by meteoric [10]Be$_m$.

**3.5 Uncertainty in the blank correction.**

The blank correction may be one of the major challenges for analyzing low [10]Be concentration samples, and a highly
variable blank could cause a scatter and increase in measured [10]Be concentrations that we observed. The blank correction
value is obtained from the average of all five blanks processed during both the high- and low-concentration sample sets.
However, the blanks are highly variable between 71,000 and 288,000 [10]Be atoms, which accounts for 10-60 % of the total
measured [10]Be atoms in the low-concentration batch. If, for sample HB-TC-02, we assume a blank of 71000 [10]Be atoms, we
get a corrected [10]Be concentration of $6.15 \times 10^5$ atoms g$^{-1}$. However, if we assume a blank of 288,000 [10]Be atoms, we get a
[10]Be concentration of $3.97 \times 10^5$ atoms g$^{-1}$, a significantly lower [10]Be concentration. Thus, variability in the measurement
background may account for a significant fraction of the difference between measured and expected concentrations. It would



only be possible to quantify this contribution of [10]Be by measuring additional blanks as well as replicates of low-concentration samples.

### 3.6 Limitations in extracting cosmogenic [10]Be from Pyroxene by fusion.

Agreement of our production rate estimate from saturated samples with all other existing data shows that extraction of cosmogenic [10]Be from pyroxene by total rapid fusion is effective and accurate for samples with high [10]Be concentrations. Previous studies of [10]Be in pyroxene used wet chemical dissolution and ion exchange chromatography, similar to the procedure used in extracting [10]Be from quartz. However, concentrations of the major cations Ca, Fe, Mg, and Na are much greater in pyroxene than the trace levels found in quartz, which requires substantial scaling up of ion exchange columns

(Eaves et al., 2018). The total fusion method of Stone (1998) , having extremely high selectivity for Be relative to these cations, completely avoids this issue. However, we were not able to sufficiently scale up the rapid fusion method to obtain the desired signal/noise ratio during AMS analysis for the lower-concentration samples.

### 3.6.1 Sample Size Limitations.

The main obstacle to measuring cosmogenic [10]Be in pyroxene at low concentrations is the difficulty in increasing the sample

size to obtain a higher [10]Be/[9]Be ratio and thus signal/background ratio. This is a challenge for both extraction methods, although for different reasons. For young exposure age samples (5-33 kyrs), Eaves et al. (2018) dissolved 1.1-2.8 g of pyroxene using large ion exchange columns. For our extraction by total fusion, the sample size is limited to 0.5 g by the size of the Pt crucibles. Note that Stone (1998) processed samples up to 4 g using 100 mL crucibles.

As discussed above, to address the crucible size limitation, we merged duplicate samples of 0.5 g to obtain a total sample mass of 1 g, but increasing the amount of K present in the final steps of the procedure most likely resulted in incomplete separation of K from Be. This, in turn, may have suppressed AMS beam currents (Table 2) and resulted in poor measurement precision for some samples. This could likely be corrected by increasing solution volumes in some steps of the procedure and repeating various precipitation steps to ensure the complete removal of K.


### Conclusion

In this study we provide advances in the measurement and application of cosmogenic [10]Be in pyroxene, by applying a rapid fusion extraction method (Stone, 1998) and a production rate calibration data set. We extracted and measured cosmogenic [10]Be in pyroxene from two sets of Ferrar Dolerite samples. One set of samples consisting of 10 high-elevation boulders

collected from moraines in the upper TAM have [3]He measurements indicating that these samples have [10]Be concentration close to saturation. We use this sample set to calibrate the production rate of [10]Be in pyroxene by assuming production-erosion equilibrium. The other set of samples consisting of 6 low-elevation glacially transported erratics from Northern



Victoria Land are used to test whether or not a rapid fusion extraction method is feasible for samples having low $^{10}$Be concentrations.


From measured $^{10}$Be concentrations in the near-saturation sample set we find the production rate of $^{10}$Be in pyroxene to be 3.74 +/- 0.10 atoms g$^{-1}$ yr$^{-1}$ which is in agreement with previously published production rate, and consistent with $^{10}$Be/$^{3}$He paired nuclide ratios from samples assumed to have simple exposure. Given the high $^{10}$Be concentration measured, a sample mass of ~0.5 g of pyroxene with 400 ug added $^{9}$Be carrier is sufficient for obtaining meaningful $^{10}$Be/$^{9}$Be ratios well above

blank levels. Even with relatively low Be yields, there is still enough total Be present for AMS detection. Therefore, the extraction of cosmogenic $^{10}$Be from pyroxene samples using rapid fusion works well for samples with high $^{10}$Be concentrations. However, for the sample set having low $^{10}$Be concentrations, the measured concentrations are higher than expected by 320,000 – 810,000 atoms g$^{-1}$. We contribute this increased $^{10}$Be concentration to potential failure in completely removing all meteoric $^{10}$Be and/or a highly variable and poorly quantified measurement background.


Advances in measuring $^{10}$Be in pyroxene and constraints on the production rate provide new opportunities for multi-nuclide measurement in pyroxene-bearing samples that allow for correcting exposure ages for surface weathering and erosion and establishing exposure-burial history.

**Code and data availability** All data information associated with the cosmogenic nuclide measurements appears in tables. The exposure age and production rate calibration in the online exposure age calculator version 3 (Balco et al., 2008) has been updated to accept data from $^{10}$Be in pyroxene.

**Author Contribution** MB carried out sample preparation for unprocessed samples. MB and LBC performed beryllium

extraction. MB and GB performed helium analysis, data reduction, and all data analysis. MB prepared the manuscript with contributions from all authors.

**Competing interests** Greg Balco is an editorial board member of Geochronology.

**Acknowledgements** We would like to thank Allie Balter-Kennedy, Shaun Eaves, and Jamey Stutz for kindly providing the samples used for this study. Further, we thank Alan Hidy of the Center for Accelerator Mass Spectrometry, Lawrence Livermore National Laboratory for Beryllium measurements. This project was supported by the U.S. National Science Foundation via grants OPP- 2139497. The LLNL portion of this work was carried out under Contract DE-AC52-07NA27344. This is LLNL-JRNL-XXXX-DRAFT.



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
