# Peer review of "Production rate calibration for cosmogenic 10Be in pyroxene by applying a rapid fusion method to 10Be-saturated samples from the Transantarctic Mountains, Antarctica."

_EGUsphere, 2024_

## Referee Comment (RC2)

**Review of manuscript egusphere-2024-702:**

*"Production rate calibration for cosmogenic $^{10}$Be in pyroxene by applying a rapid fusion method to $^{10}$Be-saturated samples from the Transantarctic Mountains, Antarctica"*

by M. Bergelin et al.

Comments by reviewer Samuel Niedermann

*General comment*

This manuscript is another contribution to various previous efforts aimed at establishing the measurement of in-situ produced cosmogenic $^{10}$Be in pyroxene as a routine tool. Here, samples from Antarctica with very high exposure ages and, thus, in presumed production-decay equilibrium for $^{10}$Be, are used to derive the $^{10}$Be production rate in pyroxene. This is an excellent way of production rate determination as it is independent of other age estimates. In addition, the authors have tested an alternative method of $^{10}$Be extraction from pyroxene (total fusion instead of wet chemical dissolution), which seems to work for high-concentration samples but less so for low-concentration samples.

Altogether, the paper is written clearly and concisely, with some exceptions as outlined below. Although it provides no final solution to the difficulties of in-situ cosmogenic $^{10}$Be extraction and determination in pyroxene, it is a valuable step forward in this direction and clearly merits being published in EGUsphere, after minor revision has taken account of the specific and technical comments given hereafter.

*Specific comments:* (numbers refer to line numbers in the manuscript)

45  Niedermann et al. (1994) is an inappropriate reference for production rate determinations based on another production rate. Although these authors compared the cosmogenic $^{21}$Ne to $^{26}$Al and $^{10}$Be in the same quartz samples, their production rate determination was indeed based on the (presumed) radiocarbon age of the studied glacially polished rock surfaces. Better references for production rate determination by comparison with another production rate are e.g. Niedermann et al. (EPSL 257, 596-608, 2007) or Luna et al. (EPSL 500, 242-253, 2018).

76  Symbols used in the equation ($N_{10}$, $P_{10}$, $\lambda_{10}$) should be explained.

149-151  What about the uncertainty of the standard? If this isn't included it should at least be mentioned here. Also, please indicate whether stated uncertainties are $1\sigma$ or $2\sigma$.

Table 3  The error shown for the $^{3}$He concentration and exposure age of 15-OTW-58 is obviously much too small, at least by a factor of 10. Other $^{3}$He errors are around 1%, which looks rather small also. Do these errors include the uncertainty of the mass spectrometer sensitivity calibration? And in case of exposure ages, what about production rate or scaling errors? If they are not included that must at least be mentioned. Also, how have uncertainties been calculated for data where there is more than one measurement per sample in the ICE-D database?

214  Also give the assumed $^{3}$He production rate!

220  "the 68% confidence interval in the measured uncertainty": Strange wording. Uncertainties are nor measured but derived from measurement statistics or propagated

from other error sources, and it's not an interval in the uncertainty but just an uncertainty or perhaps uncertainty range.

247 Two other scaling models have just been mentioned, but the only production rate value given is for St scaling. What production rate values would be obtained for Lm or LSDn scaling?

276-277 What is meant by measurement background? Only later in the manuscript it becomes clear that this is about blanks; please use clear and consistent terms.

Fig. 4 Similar issue, what is called "measured concentrations" here has been called "blank-corrected concentrations" in Table 2. There, the "measured concentratins" are not blank-corrected. Such inconsistent wording is confusing. Also, are uncertainties $1\sigma$ or $2\sigma$? $2\sigma$ would be consistent with the confidence bound of the regression line.

293-296 and Fig. 5 Please give a clear definition of the "normalized residual". Normalized to what? And what are the units of the y axis in Fig. 5? Percent?

302-303 Here again, you need to give a better explanation of what you did. What are the "replicated samples"? The low concentration samples shown in Table 4? And did you assume that the measurement with the least $^{10}$Be was free of meteoric $^{10}$Be?

314-323 If blanks are so much variable, the blank correction should take account of the whole variation, which is achieved by assuming realistic error limits. This will of course increase the uncertainty of the blank-corrected $^{10}$Be concentrations. Anyway, if variable blanks are the reason for variable $^{10}$Be concentrations I would expect values that are both too high and too low. So I doubt this can explain why $^{10}$Be concentrations are mostly higher than expected. Again, take care not to confuse blank and background (again in line 364).

*Technical comments:* (numbers refer to line numbers in the manuscript)

46 The sentence starting in line 43 is not correctly continued here: "… can be quantified in samples by … (iii) samples experiencing …". Please adjust.

Tab. 1 Gl. = Glacier?

95, 108 etc. Please use consistent style of upper case vs. lower case in section / subsection headings.

101 Balter-Kennedy et al. (2023) is missing in reference list.

119-120 Looks like an unnecessary repetition from preceding paragraph.

Tab. 2, 4 Use consistent style of upper case vs. lower case in table column headings. Please give the same number of decimal places for values and corresponding errors, also when the last decimal place is a zero. Otherwise it looks like the precision is lower

201 Stone (2000) is missing in reference list.

202 According to Table 3 this range is 3.6 – 4.3 at/g/yr.

Fig. 2 The unit yr given on the x axis is nonsense!

227 (and elsewhere) Units such as yr for years should not be given in plural form, i.e. use kyr, not kyrs.

266   Refer to Fig. 4 here.

270   Balco (2020) is missing in reference list.

References: Please use superscripts where applicable.

390   What is this? Does it belong to Balco (2016)? There is already a title there.

427   This is an incomplete reference (journal/book/publisher?) and the title contains many typos.

I have also suggested some small text corrections in the pdf file of the manuscript.

[Figure]

[Figure]

[revised manuscript text omitted]
| 15-ROB-27 | $5.92 \pm 0.11$ | $9.05 \pm 0.10$ | $8.265 \pm 0.094$ | 3.69 | Balter-Kennedy et al., (2020) |
| 15-ROB-30 | $6.78 \pm 0.14$ | $12.21 \pm 0.35$ | $9.95 \pm 0.29$ | 3.78 | Balter-Kennedy et al., (2020) |
| 15-ROB-31 | $6.40 \pm 0.12$ | $10.51 \pm 0.14$ | $8.67 \pm 0.12$ | 3.62 | Balter-Kennedy et al., (2020) |
| 15-OTW-50 | $6.27 \pm 0.13$ | $10.84 \pm 0.26$ | $9.40 \pm 0.23$ | 3.68 | ICE-D Database[b] |
| 15-OTW-55 | $6.22 \pm 0.13$ | $11.07 \pm 0.13$ | $9.56 \pm 0.11$ | 3.64 | ICE-D Database[c] |
| 15-OTW-56 | $6.65 \pm 0.14$ | $10.53 \pm 0.13$ | $9.14 \pm 0.12$ | 3.92 | ICE-D Database[c] |
| 15-OTW-57 | $6.45 \pm 0.14$ | $10.87 \pm 0.16$ | $9.28 \pm 0.13$ | 3.74 | ICE-D Database[c] |
| 15-OTW-58 | $7.67 \pm 0.17$ | $12.4235 \pm 0.0092$ | $9.0549 \pm 0.0067$ | 3.88 | ICE-D Database[c] |
| 15-OTW-60 | $7.39 \pm 0.16$ | $11.73 \pm 0.23$ | $8.54 \pm 0.17$ | 3.74 | ICE-D Database[c] |

[revised manuscript text omitted]

---

## Author Response (AR1)

**Authors' response: Production rate calibration for cosmogenic $^{10}$Be in pyroxene by applying a rapid fusion method to $^{10}$Be-saturated samples from the Transantarctic Mountains, Antarctica.**

We listed several proposed changes to the paper in our responses to RC 1 and RC 2. We have made these changes in the revised text and here we identify their locations. Note that the line numbers for the main text refer to the revised version. In addition, we have also corrected several typographical errors that we discovered in the main text as well as included the reference for previously unpublished data (line 85 and Table 1).

RC 1:

1. Add a discussion on the compositional dependence of pyroxene on production rate.
   - In section 3.2, lines 568 – 581, we added the results from major elemental composition analysis of the pyroxene sample in Table 4 and included a discussion on the compositional dependence on the production rate as suggested by RC1.
   - In summary, the range of pyroxene composition observed in the Ferrar Dolerite and previous calibration studies falls within the predicted Ferrar pyroxene composition< 6.5 %.

2. Clarification of the analytical methodology and discussions where needed
   - We have clarified some of the methodological text in sections 2.3 and 2.3 as noted in the review.

3. Include supplementary files that contain formatted input data that can easily be pasted into the online production rate calibration.
   - We have included this in the supplementary.

4. De-emphasize the statement in line 236 and the caption in Figure 3.
   - We have removed this statement from the text (line 334) and caption in Figure 3.

5. Errors and technical corrections as noted in the review.
   - We have corrected all errors throughout the main text as noted by the reviewer.

6. Update figures
   - Figure 1 has been updated to clearly distinguish between the 'measured' and 'corrected' data points for the outlier
   - Figure 3b has been updated to clearly distinguish data points from this study (circles) from that of others (triangles).

RC 2:

1. Minor specific and technical correction
   - These have been corrected throughout the main text.
2. The following tables have been updated
   - Significant figures have been updated in Tables 1, 2, 3 and 5
   - Table 3: Errors have been updated to include internal uncertainty.